# Molecular and Polymer Ln_2_M_2_ (Ln = Eu, Gd, Tb, Dy; M = Zn, Cd) Complexes with Pentafluorobenzoate Anions: The Role of Temperature and Stacking Effects in the Structure; Magnetic and Luminescent Properties

**DOI:** 10.3390/ma13245689

**Published:** 2020-12-13

**Authors:** Maxim A. Shmelev, Mikhail A. Kiskin, Julia K. Voronina, Konstantin A. Babeshkin, Nikolay N. Efimov, Evgenia A. Varaksina, Vladislav M. Korshunov, Ilya V. Taydakov, Natalia V. Gogoleva, Alexey A. Sidorov, Igor L. Eremenko

**Affiliations:** 1N. S. Kurnakov Institute of General and Inorganic Chemistry, Russian Academy of Sciences, 119991 Moscow, Russia; shmelevma@yandex.ru (M.A.S.); juliavoronina@mail.ru (J.K.V.); bkonstantan@yandex.ru (K.A.B.); nnefimov@yandex.ru (N.N.E.); judiz@rambler.ru (N.V.G.); sidorov@igic.ras.ru (A.A.S.); ilerem@igic.ras.ru (I.L.E.); 2P. N. Lebedev Physical Institute, Russian Academy of Sciences, 119991 Moscow, Russia; janiy92@yandex.ru (E.A.V.); vladkorshunov@bk.ru (V.M.K.); taidakov@mail.ru (I.V.T.); 3Faculty of Fundamental Sciences, Bauman Moscow State Technical University, 105005 Moscow, Russia; 4Academic Department of Innovational Materials and Technologies Chemistry, Plekhanov Russian University of Economics, 117997 Moscow, Russia

**Keywords:** cadmium-lanthanide(III) complexes, zinc-lanthanide(III) complexes, pentafluorobenzoic acid, coordination polymer, single crystal X-ray, magnetochemistry, photoluminescence

## Abstract

Varying the temperature of the reaction of [{Cd(pfb)(H_2_O)_4_}^+^*_n_*·*n*(pfb)^−^], [Ln_2_(pfb)_6_(H_2_O)_8_]·H_2_O (Hpfb = pentafluorobenzoic acid), and 1,10-phenanthroline (phen) in MeCN followed by crystallization resulted in the isolation of two type of products: 1D-polymers [LnCd(pfb)_5_(phen)]*_n_***·**1.5*n*MeCN (Ln = Eu (I), Gd (II), Tb (III), Dy (IV)) which were isolated at 25 °C, and molecular compounds [Tb_2_Cd_2_(pfb)_10_(phen)_2_] (V) formed at 75 °C. The transition from a molecular to a polymer structure becomes possible because of intra- and intermolecular interactions between the aromatic cycles of phen and pfb from neighboring tetranuclear Ln_2_Cd_2_ fragments. Replacement of cadmium with zinc in the reaction resulted in molecular compounds Ln_2_Zn_2_ [Ln_2_Zn_2_(pfb)_10_(phen)_2_]·4MeCN (Ln = Eu (VI), Tb (VIII), Dy (IX)) and [Gd_2_Zn_2_(pfb)_10_(H_2_O)_2_(phen)_2_]·4MeCN (VII). A new molecular EuCd complex [Eu_2_Cd_2_(pfb)_10_(phen)_4_]·4MeCN (X)] was isolated from a mixture of cadmium, zinc, and europium pentafluorobenzoates (Cd:Zn:Ln = 1:1:2). Complexes II-IV, VII and IX exhibit magnetic relaxation at liquid helium temperatures in nonzero magnetic fields. Luminescent studies revealed a bright luminescence of complexes with europium(III) and terbium(III) ions.

## 1. Introduction

Luminescent lanthanide complexes attracted much attention because of their potential applications in biomedical imaging, catalysis [1], quantum computing technics [2], magnetic refrigerators [3], sorbents [4], luminescent sensors, and fluorescent probes [5]. The narrow bandwidth of the f-f transitions, the long luminescence lifetime (~ms) and the presence of high magnetic anisotropy determine the interest in the studies being carried out and the prospects for practical applications. The use of aromatic ligands in the design of complexes of luminescent lanthanides is required to enhance their emission. Carboxylate ligands were studied as efficient sensitizers for luminescent lanthanide complexes as well as molecular magnets [6,7]. Fluorinated organic ligands are actively used to improve the intensity of luminescence of complexes by reducing the effect of fluorescence quenching by the vibrational C-H bond [8,9,10,11,12]. For example, Eu(III) perfluorobenzoates reveal much higher luminescence than benzoates [13]. Terbium(III) and europium(III) fluorobenzoates were shown to possess high luminescence intensity with quantum yields up to 70% [14]. The use of co-ligands and/or d-metals in the design of lanthanide complexes leads to structural modification needed to improve their luminescent and/or magnetic properties due to additional antenna effect, control the geometry of the coordination polyhedra of lanthanide atoms, minimize interionic interactions, etc. [15,16,17,18,19,20,21,22]. Fluorinated polycarboxylic acids are used for the synthesis of coordination polymers (CPs) as well as metal organic frameworks (MOFs). Fluorinated lanthanide MOFs/CPs are of great interest for their applications as OLED (organic light-emitting diode) sensors, in biomedical analysis and in cell imaging [12,14,23,24,25,26].

A special role in the formation of complexes with anions of fluorinated carboxylic acids and aromatic N-donor ligands is played by π-π interactions between the aromatic fragments of heteroligans, which additionally stabilize the crystal structure [27,28,29]. Additionally, fluorinated carboxylic acids are involved in the formation of C-H...F and C...F, F...F contacts, which also affect the stability of the crystal packing of the complexes [5,30,31,32]. These intra- and intermolecular interactions can positively affect the luminescent properties of the resulting complexes [33,34,35].

This work presents the results on the synthesis of LnZn and LnCd complexes with pentafluorobenzoate anions (Hpfb) and 1,10-phenantroline (phen) molecules. In spite of the fact that studies of heterometallic complexes are attracting more and more attention in the search for ways to modify physicochemical properties, this area of coordination chemistry remains scarcely studied. A series of hitherto unknown molecular and polymeric LnZn and LnCd complexes were obtained and the conditions for the conversion of the molecular form of a LnCd complex to the corresponding 1D polymeric one without changing the compound composition were determined. The magnetic and luminescent properties were compared systematically.

## 2. Results

### 2.1. Synthesis of Complexes

The reaction between [{Cd(pfb)(H_2_O)_4_}^+^*_n_*·*n*(pfb)^−^] [28] and [Ln_2_(pfb)_6_(H_2_O)_8_]·2H_2_O [13,36] in the presence of phen (in the 2:1: 2 ratio, respectively) in MeCN at room temperature gave a series of 1D-coordination polymers with the composition [LnCd(pfb)_5_(phen)]*_n_*·1.5*n*MeCN (Ln = Eu (I), Gd (II), Tb (III), Dy (IV)) (yield ~65–80%). It was found that increasing the synthesis temperature to 75 °C and crystallization of III in a sealed vial allowed us to isolate the crystals of [Tb_2_Cd_2_(pfb)_10_(phen)_2_] (V) complex in ~40% yield. The composition of V is similar to that of polymer III, except for the presence or absence of solvate molecules. Compound V is a molecular complex, as confirmed by X-ray diffraction data. Isolation of molecular complexes with Eu^3+^, Gd^3+^, and Dy^3+^ cations was delayed by a search for the temperature conditions for reproducible isolation of single crystals as single-phase products.

On replacement of the cadmium salt with the zinc one in reactions for the synthesis of I–V and at the synthesis and crystallization temperatures varied from 25 °C to 80 °C, only tetranuclear complexes with the composition [Ln_2_Zn_2_(pfb)_10_(phen)_2_]·4MeCN (Ln = Eu (VI), Tb (VIII), Dy (IX)) and [Gd_2_Zn_2_(pfb)_10_(H_2_O)_2_(phen)_2_]·4MeCN (VII) were precipitated in a good yield (60–80%).

Our attempt to synthesize a heterometallic {LnZnCd} compound using the above reagents in the ratio Zn:Cd:Eu:phen = 1:1:1:2 resulted in the crystallization of a new molecular EuCd heterometallic complex [Eu_2_Cd_2_(pfb)_10_(phen)_4_]·4MeCN (X) in about 20% yield.

The resulting compounds were obtained as single-crystal and polycrystalline solids. The purity of the bulky obtained samples was proven by PXRD (see supporting information, Appendix A). All the new compounds were characterized by C, H, N-analysis and IR spectroscopy. The equivalence of their compositions with the molecular formulas was confirmed by X-ray diffraction studies.

Isostructural compounds I–IV were crystallized as single-crystal solvates with acetonitrile; they were found to consist of polymer chains of alternating pairs of Cd and Ln atoms, which form tetranuclear Ln_2_Cd_2_ fragments (Figure 1). The chain growth occurs via bridging and chelate-bridging carboxylate anions. The coordination polyhedron LnO_9_ is a muffin constructed by oxygen atoms of three bridging, three chelate-bridging, and one bidentate-coordinated carboxylate anions (Table 1 and Appendix A). The coordination polyhedron CdO_5_N_2_ is a pentagonal bipyramid formed by three chelate-bridging and one bridging carboxylate anion and completed by the coordination of two nitrogen atoms of the phenanthroline molecule (Table 1 and Appendix A). The elementary chain unit is a fragment that contains one Cd atom and one Ln atom; the chain growths occurs through the center of inversion. As a result, a metal core is formed in which the cadmium and lanthanum atoms are located in different planes. The angle between these planes is about 80°, and the aromatic fragments of the ligands are arranged symmetrically in opposite quarters formed by the intersection of these planes. The aromatic nature of all substituents and their almost strictly perpendicular arrangement results in a significant amount of π...π overlaps throughout the polymer chain. The geometric parameters of these interactions (the distances between the planes of the cycles are 3.2–3.4 Å, the angles between these planes are 4–6°, Appendix A) indicate that they have high strength and make a large contribution to the formation of the structure of the polymers studied because of there are very many of them.

The polymer chains in the crystal are located parallel to the crystallographic axis *a* and are bonded by C-F…π and C-H…F interactions (Appendix A). The acetonitrile molecules are embedded in the crystal lattice due to C-H...N and C-N...π interactions (Appendix A).

Compound V is a molecular tetranuclear complex consisting of two Ln atoms in the center of the molecule and two cadmium atoms on the periphery (Figure 2). The chemical composition of the complex is identical to that of polymer III; therefore, it is of particular interest to compare the structures of molecular complex V and the analogous tetranuclear fragment in III. The metal atoms in compound V are in the same plane, which results in an increase in the Cd-Tb-Tb angle (115.23(3)° and 171.62(1)° in III and V, respectively) and the distance between the cadmium atoms in comparison with compound III (10.463(4) Å in III and 11.822(1) Å in V). The coordination polyhedron TbO_8_ is a triangular dodecahedron. The geometry of the CdO_5_N_2_ polyhedron is a pentagonal bipyramid (Appendix A). A detailed analysis of the geometric parameters and the overlap of the corresponding fragments showed that the central fragment of two lanthanides bound by pentafluorobenzoic anions is identical in the structures of III and V; the polymer and molecular complexes differ by the structures of the Ln-Cd fragments. This is due to the mutual arrangement of the aromatic fragments of the ligands. In V, the C_6_F_5_ fragments are located quite freely and the intramolecular π...π overlaps are weak (Appendix A); the phenanthroline molecules “close” the complex and are located practically perpendicular to the planes of the C_6_F_5_ rings. The large aromatic fragments of phen of neighboring molecules in a crystal of V overlap each other, participating in a strong π...π interaction and forming the main supramolecular motif of the crystal structure of this compound—infinite chains located along the main diagonal of the crystal (Appendix A). Due to C-F...π, C-H…O, and C-H...F interactions, the chains are stacked parallel to each other to form a three-dimensional crystal structure (Appendix A).

Compounds VI–IX are zinc analogs of molecular complex V. The geometry of the “central” lanthanide fragment in them is the same as in V (Figure 3). In addition, the geometry of the Ln-Zn fragment slightly differs due to a change in the type of binding of pfb-anions. First of all, the changes in the structure of the complexes appear in lanthanide polyhedra: in isostructural compounds VI, VIII, and IX, three different polyhedrons have the same coordination number equal to that in VIII, while in the structure of VII there is an additional coordinated water molecule; as a result, the changes are most significant (Table 2 and Appendix A). The heterometallic atoms in VI—IX are bound by one chelate-bridging and two bridging pfb-groups (the geometry of the ZnO_6_N_2_ polyhedron corresponds to a distorted octahedron (Appendix A)); as a result, there is a redistribution of longer and shorter bonds, which results in a change in the relative positions of the C_6_F_5_ rings and π...π interactions involving these rings (Appendix A).

In addition, the phenanthroline fragments of neighboring molecules in the crystals of these compounds are located slightly more closely than in cadmium complexes, which indicates a slightly higher energy of their overlapping. In general, the crystal structure of zinc complexes is similar to that of cadmium complexes, i.e., parallel chains of molecules linked by weak interactions into a three-dimensional lattice.

The structure of complex VII differs from those of VI, VIII, and IX. It has an additional water molecule coordinated by the gadolinium atom. This leads both to a change in the polyhedron of the corresponding atom (capped square antiprism) and to changes in the structure of the complex (Appendix A). The Ln-Ln distances are slightly elongated while the Zn-Ln distances are shortened (Table 2). The appearance of a hydrogen bonding center―a water molecule―in the structure of VII promotes H-bonding of MeCN solvate molecules via classical H-bonds (Appendix A).

The packing of crystals VI–IX is realized due to π...π, C-F...π, C-H…O and C-H...F interactions (Appendix A).

Compound X (Figure 4, Table 1) is a molecular complex similar to V. The difference is that the place of one pfb-ligand is occupied by an additional phenanthroline molecule coordinated by the Eu1 atom. The carboxylate anions change their character from chelate-bridging to bridging in the central Eu2 fragment, thereby freeing up one coordination place on the Eu1 atom, while the chelate-bridging pfb-ligand in the LnCd fragment performs a chelating function on the Cd1 atom. The phenanthroline molecules coordinated by the Cd1 and Eu1 atoms are located in parallel planes, almost perpendicular to the plane of the metal atoms. They participate in an intramolecular π...π interaction the geometric parameters of which allow us to assume its significant strength (Appendix A). The peripheral ligand in this linear compound is a chelated pfb ligand. The molecular packaging in the crystal is a zigzag chain parallel to the *a* axis formed by overlapping of the phenanthroline fragments of the neighboring molecules. The further development of crystal packaging is due to weaker interactions (Appendix A).

Thus, our analysis of the molecular and crystal structure of compounds I–X has shown that the fragment of two lanthanide atoms and four pentafluorobenzoate fragments has the same structure regardless of its environment. Only the angle of the pentafluorbenzene rings rotation differs because of the high mobility of C-C bonds. It is also evident in the disordering of these fragments in the resulting structures, and has no effect on the geometry of the metal core. It has also been shown that the main structure-forming non-covalent interactions in the crystals studied are interactions involving the π systems of pentafluorobenzoate and phenanthroline ligands. These interactions determine both the molecular and crystal structure of the studied complexes.

### 2.2. Photoluminescence

The luminescence properties of the solid-state europium, terbium and gadolinium compounds I–III and V–VIII were studied in detail. The room temperature luminescence spectra of europium (Figure 5) and terbium (Figure 6) samples exhibit narrow f-f bands of the corresponding lanthanide ions as well as weak broad emission bands of *d*-block that are typical of heteronuclear lanthanide compounds (Figure 7) [37,38,39].

The narrow emission bands of complexes I and VI correspond to the ^5^D_0_→^7^F*_J_* (*J* = 0–4) transitions of Eu^3+^ (Figure 5). The integrated intensity ratios of ^5^D_0_→^7^F_2_ hypersensitive transition to the magnetic dipole transition ^5^D_0_→^7^F_1_ of europium compounds I and VI are 4.06 and 5.97, respectively, and indicate a deviation of the Eu^3+^ site symmetry from the inversion center. The ^5^D_0_→^7^F_0_ transition displays a single symmetrical narrow component in all cases that indicates the presence of only one type of the Eu^3+^ environment. This is confirmed by the fact that all luminescence decay curves of the excited ^5^D_0_ level of Eu^3+^ and of the excited ^5^D_4_ level of Tb^3+^ were well fit by a monoexponential decay function (Table 3).

The f-f luminescence bands of III, V and VIII are assigned to ^5^D_4_→^7^F*_J_* (*J* = 6–3) transitions of Tb^3+^ (Figure 6). The emission spectra of all terbium compounds are dominated by the ^5^D_4_→^7^F_5_ transition at 545 nm responsible for the green color of luminescence. The similarity of the Stark splitting structure of the electronic transitions is indicative of the identity of the Tb^3+^ ion’s nearest environment in Cd-containing compounds and insignificant differences in the Zn-containing compound.

The broad-band luminescence of *d*-block in the 340–450 nm wavelength range with a vibrational fine structure is clearly observed in emission spectra of europium, terbium, and gadolinium compounds (Figure 7). The existence of these bands in the luminescence spectra indicates an incomplete energy transfer from the *d*-block to the lanthanide ions.

The excitation spectra of I, III, V, VI and VIII recorded at room temperature are displayed in Figure 8. In addition to the S_0_–S_1_ transition of phen ligand with maximum at ~29,000 cm^−1^ and f-f transitions of the lanthanide ions the excitation spectra of molecular compounds V, VI and VIII display wide shoulders located up to 26,000 cm^−1^. As the compounds characterized by the π-stacking interaction, the shoulder can be assigned to the interligand charge transfer (ILCT state) as the result of ligand charge redistribution. Moreover, the intense shoulder extended up to 24,500 cm^−1^ of the europium compound VI is absent in isostructural terbium complex VIII and therefore indicates participation of low-lying ligand-to-metal charge transfer state (LMCT state) in the energy transfer processes of VI. It is noteworthy that compared with the molecular terbium compound V the energy of the first excited singlet state of the polymer III shows remarkable blue shift by about 300 cm^−1^. The reason can also be π-staking leading to bathochromic shift of the excitation maximum [11]. The domination of the ligand absorption bands in the excitation spectra proves effective luminescence sensitization via excitation of the ligand.

The triplet state energy of the d-block was determined from the 77 K phosphorescence spectra of Gd compounds II and VII (Appendix A) using the short-wavelength edge corresponding to 0–0 phonon transition. So, the triplet energy of d-block is 21,500 cm^−1^, which correlates well with the reported value of the triplet state energy of phen ligand [11]. The energy differences between the S_1_ and T_1_ states is about 7500 cm^−1^, which exceeds the preferred value (5000 cm^−1^ [40]) providing the most efficient intersystem crossing. However, the charge transfer states described above can promote an increase in sensitization efficiency by acting as intermediate states in the energy transfer process.

The long lifetimes (Table 3) clearly reflect a lanthanide environment devoid of water molecule. The considered quantum yields of the compounds are quite similar with the exception of terbium-containing polymer III that demonstrates a higher luminescence efficiency as well as longer lifetime of the Tb^3+^ excited state ^5^D_4_. Therefore, the polymerization of III results in almost twice higher luminescence quantum yield compared to its molecular analogue V.

The long lifetimes (Table 3) clearly reflect a lanthanide environment devoid of a water molecule. The quantum yields of the compounds considered are quite similar, with the exception of terbium-containing polymer III that demonstrates a higher luminescence efficiency as well as longer lifetime of the Tb^3+^ excited state ^5^D_4_. Therefore, the polymer structure of III results in an almost two times higher luminescence quantum yield compared to its molecular analogue V.

The efficiency of the energy transfer processes in the europium complexes was estimated by calculating the intrinsic quantum yields via Werts’ formula [41]. The data obtained are reported in Table 3. The luminescence efficiency is essentially limited by an incomplete energy transfer from the d-block to the emission center. Nevertheless, the quantum yield of I and VI is much higher than that of homonuclear europium compounds with pfb ligands: the reported quantum yield is 15% [13]. The presence of the phen ligand result in an increase in sensitization efficiency due to more favorable energies of its excited states. Moreover, the absence of high-frequency oscillating bonds achieved by the presence of a *d*-block provide a significantly longer lifetime of the europium excited state regardless of the presence of phen as a ligand [5]. Since terbium luminescence is less susceptible to quenching due to a larger energy gap between the lowest excited state and the highest ground state levels, the lifetime difference between heteronuclear and homonuclear terbium compounds is less significant [42].

### 2.3. Magnetic Properties

The magnetic properties of complexes II–IV and VII–IX were studied in the temperature range of 2–300 K in a 5000 Oe dc-magnetic field (Figure 9; the χ_M_T values for II-IV were calculated in terms of the Ln_2_Cd_2_ unit for clarity). The M(H) and M(H/T) plots for II-IV and VII-IX are shown in Appendix A. The χ_M_T(T) plots are similar for complexes with identical lanthanides ions. On cooling from room temperature to 2 K, the χ_M_T values for II–IV, VII–IX remain nearly constant up to 9 K (II, VII) or 150 K (II, IV, VIII, IX). On further cooling, first a slow decrease and then a sharp drop in χ_M_T followed by a decrease to minimum values at 2 K occurs (Figure 9). The most indicative χ_M_T values are presented in Table 4, along with theoretical ones for the corresponding isolated Ln^3+^ ions. The experimental χ_M_T values for complexes II–IV and VII–IX (see Table 4) are in good agreement with the theoretical ones.

With an eye to evaluate the magnitude of magnetic interactions in II and VII, the temperature dependences of χ_M_T were fitted (Table 5, Appendix A, Figure 9, Appendix A) using the PHI program developed by Chilton et al. [44] using$\;\hat{H}=J{\hat{S}}_{1}{\hat{S}}_{2}+\mu_{B}\mathrm{gB}\left(\hat{S}1+\hat{S}2\right)\;$Hamiltonian, where ${\hat{S}}_{1}=\;{\hat{S}}_{2}={\hat{S}}_{Gd}$, J is the intramolecular magnetic exchange constant, g is the g-factor, B is the magnetic flux density, and μ_B_ is the Bohr magneton.

Intermolecular interactions (zJ) between Gd atoms were also considered using the mean-field approximation:$$\chi_{zJ}=\;\frac{\chi_{calc}}{1-\left(\frac{zJ}{N_{A}\mu_{B}^{2}}\right)\chi_{calc}}$$
where *χ_zJ_* is the refined magnetic susceptibility based on intermolecular interaction, *χ_calc_* is the magnetic susceptibility calculated on the basis of the Hamiltonian, *N_A_* is the Avogadro number, and *μ_B_* is the Bohr magneton. The calculation errors were estimated as
$$R=\;\overset{n}{\underset{i=1}{\sum }}{\left(\chi_{exp}-\chi_{calc}\right)}^{2}$$
where *n* is a number of experimental points.

According to the values of intra- and intermolecular interactions presented in Table 5, one can assume that the magnetic interaction between the Gd^3+^ ions is relatively weak and cannot considerably change the magnetic behavior of complexes II and VII. The values of the exchange coupling parameters are comparable with those observed for binuclear gadolinium complexes with a similar structure [45,46].

Slow relaxation of magnetization, which has a number of promising practical applications, is often observed in lanthanide complexes, especially in Dy-containing ones. In order to study the magnetic behavior dynamics of the complexes, the magnetic susceptibility in an alternating magnetic field (ac-magnetic susceptibility) was investigated. In zero dc-magnetic field, a significant signal on the frequency plots of the imaginary component (out-of-phase signal) of ac-magnetic susceptibility χ^″^(ν) was observed only for complexes containing Dy (IV and IX); however, the peak is out of the frequency range of the equipment used (10–10,000 Hz). Therefore, the evaluation of the relaxation characteristics of these complexes does not seem possible. For all the other complexes, the deviations from zero in the χ^″^(ν) plots are within the measurement error of the magnetometer. Such a behavior is often associated with the dominant influence of the quantum tunneling of magnetization (QTM) relaxation process. The QTM significantly accelerates relaxation, making it unobservable. In order to suppress this relaxation process for complexes II-IV and VII-IX, the ac-magnetic susceptibility was studied in dc-magnetic fields of up to 5000 Oe at 2 K. The latter allowed us to obtain the frequency dependences of the real (χ’) and imaginary (χ^″^) components of the ac-magnetic susceptibility (Appendix A).

Relaxation deceleration of compound VIII is not observed at any dc-magnetic field applied (Appendix A). For complexes III and IV, despite a noticeable decrease in the magnetic relaxation rate, the maxima on the χ^″^(ν) curves are outside the frequency range of the equipment used at any dc-magnetic fields (Appendix A), which does not allow determining the parameters of the relaxation processes of these compounds.

Varying the external dc-magnetic field strength made it possible to obtain the optimal values which correspond to the slowest relaxation for complexes II (1000 Oe), VII (2500 Oe) and IX (1500 Oe) (Appendix A). By applying optimal fields, the frequency dependences of the real (χ′) and imaginary (χ^″^) components of the ac-magnetic susceptibility (Figure 10, Appendix A) were obtained in different temperature ranges. Based on these results, the χ^″^(ν) isotherms were approximated using the generalized Debye model and dependences of the relaxation time on the reciprocal temperature τ(1/T) for compounds II, VII, IX (Figure 11, Appendix A) were obtained.

It is important to note that an increase in the intensity of χ^″^(ν) signal was observed for IX with a rise in temperature from 2 K to 3 K (Figure 10). This pattern may be due to the collective behavior caused by the weak dipole-dipole or exchange interactions between the Ln^3+^ ions [3,47,48,49,50,51,52].

To determine the parameters of the relaxation processes of the compounds, the high-temperature part of the τ (1/T) dependences were approximated by the Orbach relaxation mechanism:
τ_Or_^−1^ = τ_0_^−1^ exp(−Δ*E*/k_B_*T*)(1)
where Δ*E*—effective energy barrier; k_B_—Boltzmann constant, τ_0_—preexponential factor, *T*—temperature.

A similar estimation of the relaxation parameters of Gd-containing complexes II and VII in the temperature ranges of 5–6 K (II) and 5.5–6.5 K (VII) gives the following values: for II—τ_0_ = 9 × 10^−6^ (± 2 × 10^−6^) s, Δ*E*/k_B_ = 4 (± 1) K; for VII—τ_0_ = 9.8 × 10^−6^ (± 5 × 10^−7^) s, Δ*E*/k_B_ = 4.1 (± 0.3) K. For compound IX, the relaxation parameters for the temperature range 6.5–7.25 K are τ_0_ = 5 × 10^−8^ (± 1 × 10^−8^) s and Δ*E*/k_B_ = 41 (± 2) K. The plots of τ(1/T) dependences for complexes II, VII and IX in a semilogarithmic coordinate system are nonlinear (Figure 11, Appendix A). This indicates the presence of relaxation mechanisms that are different from the Orbach one.

It should be borne in mind that the Orbach relaxation mechanism often does not contribute to relaxation, as previously observed in a number of examples for Er and Yb complexes [52]. The latter is suggested by the τ_0_ values and also by approximation of the τ(1/T) plot by the sum of the Raman and QTM relaxation mechanisms (τ^−1^ = C_Raman_T^nRaman^ + B; for complexes II and VII) or Raman and direct relaxation mechanisms (τ^−1^ = C_Raman_T^nRaman^ + A_direct_TH^4ndirect^; for complex IX) over the entire temperature range. The relaxation times characteristic of over-barrier magnetization reversal corresponding to the Orbach mechanism should be ~10^−10^–10^−12^ s [53]. The τ_0_ parameters of complexes II, VII, IX differ significantly from the indicated values. By using a sum of the QTM and Raman mechanisms in the approximation of the experimental data (Figure 11, Appendix A), the following relaxation parameters were obtained: for II—C_Raman_ = 2143 (± 508) K^−n-Raman^ s^−1^, n_Raman_ = 1.6 (± 0.1), B = 24,360 (± 1149) s^−1^; for VII, C_Raman_ = 731 (± 499) K^−n-Raman^ s^−1^, n_Raman_ = 2.2 (± 0.4 B = 15,938 (± 1973) s^−1^; for complex IX**,** the optimal approximation was achieved using the sum of the Raman and direct relaxation mechanisms with parameters: C_Ram_ = 0.6 (± 0.2) K^−n-Raman^ s^−1^, n_Raman_ = 5.8 (± 0.2), A_direct_ = 2.26 × 10^−11^ (± 5 × 10^−13^) K^−^^1^Oe^−^^4^s^−^^1^, n_direct_ = 4 (fixed for Kramers ions). The use of other sets of the mechanisms than those presented above for approximation results in over-parameterization.

## 3. Discussion

The fact that polymer complexes I–IV formed was found to be somewhat unexpected, since the formation of molecular structures is typical of M-Ln (where M is a transition metal) compounds with monocarboxylic acid anions and phen molecules or its substituted analogs (based on the data of CCDC (Nov. 2019 + 3 upd) [54] which describes 57 molecular compounds). Probably, the stabilization of the polymer structure is associated with non-covalent interactions in the crystal packing of the complexes, which causes the arrangement of pentafluorophenyl substituents of carboxylate anions and coordinated phen molecules to be close to parallel. Due to this mutual orientation of aromatic fragments, the carboxylate ligand coordinated by the Cd atom participates in the formation of π-π interactions with the phen molecule of the neighboring Ln_2_Cd_2_ fragment, while the O atom in its composition performs the μ_2_-bridging function between two cadmium atoms.

A similar bonding of mononuclear fragments accompanied by the mutual orientation of aromatic rings was previously observed for cadmium pentafluorobenzoate complexes [Cd(H_2_O)(bpy)(pfb)_2_]*_n_* (where bpy–2,2′-bipyridine) [32], [Cd(phen)(pfb)_2_]*_n_* [55], and [Eu_2_(H_2_O)_2_Cd(etpy)(pfb)_8_]*_n_* (where etpy–3-ethinylpyridine) [28]. As we have already noted, we were unable to isolate the pure phase of molecular complex V or its analogs comprising other lanthanides. This is an interesting case for comparing the photophysical characteristics of molecular and polymeric compounds with the same composition. In addition, the availability of a sufficient amount of a pure sample of molecular complex V would make it possible to study the possibility of transition of V to III in a thermolysis process, taking DSC and XRD data into account.

Magnetic measurements showed that gadolinium (II and VII) and dysprosium (IX) complexes behave as field-induced SMM. The gadolinium(III) complexes can exhibit magnetic relaxation [56,57,58,59,60] either due to weak anisotropy [56] or a low rate of relaxation processes other than Orbach relaxation. The observed effect requires more thorough studies. Dysprosium-containing binuclear carboxylates were previously discussed [61] with cinnamate derivatives [Dy_2_(L)_6_(DMSO)_2_(H_2_O)_2_] (where L is 3-methoxycinnamate or 2-methoxycinnamate) with coordination geometry of Dy atoms (DyO_9_) described as a capped square antiprism as examples. They relax through a combination of the Raman and direct processes; the Orbach relaxation is also involved but was found to be negligible. Tuning of the coordination environment of the dysprosium atom with N-donor chelate ligands (phen or bpy) in binuclear 4-chlorobenzoates (DyO_6_N_2_–square antiprism) [62] or phenoxyacetate (DyO_7_N_2_–muffin) [63] led to the predominance of the Orbach relaxation process in magnetic relaxation for the former. The geometry of the DyO_9_ polyhedron in the complexes described above corresponds to a muffin or a triangular dodecahedron. These results show an essential role of the coordination geometry of Dy(III) ions (preference is given to a square antiprism) in determining the type of magnetic relaxation.

## 4. Experimental

### 4.1. Materials and Methods

The new compounds were synthesized in the air using acetonitrile (MeCN, C. P.) as solvents. Following reagents were used as received in synthetic procedures: Eu(NO_3_)_3_·6H_2_O (99.99%, «Lanhit»), Tb(NO_3_)_3_·6H_2_O (99.99%, «Lanhit»), Gd(NO_3_)_3_·6H_2_O (99.99%, «Lanhit»), Dy(NO_3_)_3_·5H_2_O (99.99%, «Lanhit»), H(pfb) (99%, «P&M Invest»), phen·H_2_O (99%, «Alfa Aesar»). [{Cd(H_2_O)_4_(pfb)}^+^*_n_·**n*(pfb)^−^] [28] and [Ln_2_(pfb)_6_(H_2_O)_8_]·2H_2_O [13,36] were synthesized according to the previously reported procedures. Zn(OH)_2_ was synthesized by the reaction of stochiometric amounts of KOH and Zn(NO_3_)_2_·6H_2_O in water.

IR spectra of the complexes were recorded using a Perkin Elmer Spectrometer 65 (PerkinElmer, Waltham, MA, USA) equipped with a Quest ATR Accessory (Specac) by means of attenuated total reflectance (ATR) in the range of 4000–400 cm^−1^. C, H, N, S-analysis of the presented complexes was performed on a EuroEA 3000 device (Eurovector, Via Fratelli Cuzio, Italy).

The excitation and emission spectra were measured with a Fluorolog FL3-22 (Horiba-Jobin-Yvon) spectrofluorimeter (Horiba Scientific, Kyoto, Japan) equipped with a xenon lamp (450 W) and a R-928 photomultiplier (Hamamatsu Photonics, Hamamatsu, Japan). The spectra were corrected for instrumental responses. Lifetimes were measured with the same instrument using a xenon flash lamp. The overall quantum yields (QY) were determined by absolute method using a Spectralone-covered G8 integration sphere (GMP SA, Renens, Switzerland) coupled to the spectrofluorimeter.

Magnetic susceptibility measurements were carried out using a Quantum Design PPMS-9 susceptometer (Quantum Design, San Diego, CA, USA). The static magnetic susceptibility was measured at a magnetic field strength of 5000 Oe in the temperature range of 2–300 K. Alternating field measurements were performed at 1, 3 and 5 Oe in a frequency range of 10–10,000 Hz. All polycrystalline samples for the magnetic measurements were placed in polyethylene packs and mixed with an inert mineral oil to preclude the orientation of the crystals in magnetic field. In magnetic susceptibility calculations, the paramagnetic components of χ were determined with respect to the sample holder and mineral oil contributions and the diamagnetic contribution of the sample estimated from Pascal’s constants.

Single crystal X-Ray diffraction experiments with I–X were done on a Bruker Apex II diffractometer (Bruker, Billerica, MA, USA) with a CCD camera and a graphite monochromated MoKα radiation source (*λ* = 0.71073 Å) [64]. Semiempirical absorption corrections were used in all the experiments [65]. Direct methods and Fourier techniques were used in structure solving. The refinement was done by the full-matrix least squares technique against F2 with anisotropic thermal parameters for all non-hydrogen atoms. The H atoms of the hydroxy groups were calculated by difference Fourier synthesis and geometrically in the other cases. The H atoms in the structures were refined in the riding model. Using Olex2, [66] the structure was solved by direct methods in the ShelXS program package. The structures were refined by ShelXL [67]. The SHAPE 2.1 software [68] was used to determine the metals polyhedrons. The most important experimental crystallographic data and refinement statistics are reported in Appendix A. Supplementary crystallographic data for the compounds synthesized are given in CCDC numbers 2035046 (for I), 2035047 (for II), 2035048 (for III), 2035049(for IV), 2035050 (for V), 2035051 (for VI), 2035052 (for VII), 2035053 (for VIII), 2035055 (for IX), 2036917 (for X). These data can be obtained free of charge from The Cambridge Crystallographic Data Centre via www.ccdc.cam.ac.uk/data_request/cif.

The PXRD data in the form of powder patterns were collected on a Bruker D8 Advance diffractometer (Bruker, Billerica, MA, USA) with a LynxEye detector in Bragg-Brentano geometry, with the sample thinly dispersed on a zero-background Si sample holder, λ(CuKα) = 1.54060 Å, θ/θ scan with variable slits (irradiated length 20 mm), 2θ from 5° to 41°, step size 0.02°.

### 4.2. Synthesis


*[LnCd(pfb)_5_(phen)]_n_·1.5nMeCN (Eu(I), Gd(II), Tb(III), Dy(IV)).*


Compound [Ln_2_(pfb)_6_(H_2_O)_8_]·2H_2_O (0.083 mmol, Ln = Eu(I), Gd(II), Tb(III), Dy(IV)) was added to a solution of [{Cd(H_2_O)_4_(pfb)}^+^*_n_*·*n*(pfb)^−^] (0.100 g, 0.166 mmol) in 5 mL MeCN. The reaction mixture was stirred at 75 °C for 20 min, then 0.029 g of phen (0.166 mmol) was added. The solution was kept in a sealed vial at room temperature. Colorless crystals suitable for X-ray diffraction studies that precipitated after 7 days were filtered off, washed with cold MeCN (T = 5 °C), and dried in air at 20 °C.

The yield of I was 0.177 g (68.9%) based on [{Cd(H_2_O)_4_(pfb)}^+^*_n_*·*n*(pfb)^−^]). Anal. Calc. for C_50_H_12.5_O_10_N_3.5_F_25_EuCd (%): C 38.5; H 0.8; N 3.1. Found(%): C 38.3; H 0.5; N 2.9. IR (ATR), ν/cm^–1^: 1649 m, 1627 m, 1589 s, 1574 m, 1518 m, 1488 s, 1449 m, 1433 m, 1384 s, 1348 m, 1283 m, 1222 w, 1136 m, 1103 s, 990 s, 930 m, 864 m, 846 m, 826 m, 759 s, 725 s, 696 s, 650 m, 639 m, 619 w, 556 w, 545 w, 504 m.

The yield of II was 0.209 g (81.4%) based on [{Cd(H_2_O)_4_(pfb)}^+^*_n_*·*n* (pfb)^−^]). Anal. Calc. for C_50_H_12.5_O_10_N_3.5_F_25_GdCd (%): C 38.3; H 0.8; N 3.1. Found(%): C 38.3; H 0.6; N 3.0. IR (ATR), ν/cm^–1^: 1649 m, 1627 m, 1589 s, 1574 m, 1518 m, 1488 s, 1449 m, 1433 m, 1384 s, 1345 m, 1284 m, 1283 w, 1221 w, 1136 m, 1103 s, 989 s, 930 m, 864 m, 846 m, 823 m, 759 s, 725 s, 697 s, 650 m, 640 m, 619 w, 556 w, 546 w, 505 m.

The yield of III was 0.200 g (78.1%) based on [{Cd(H_2_O)_4_(pfb)}^+^*_n_*·*n* (pfb)^−^]). Anal. Calc. for C_50_H_12.5_O_10_N_3.5_F_25_TbCd (%): C 38.3; H 0.8; N 3.1. Found(%): C 38.4; H 0.7; N 3.3. IR (ATR), ν/cm^–1^: 1649 m, 1627 m, 1589 s, 1574 m, 1518 m, 1488 s, 1449 m, 1433 m, 1384 s, 1348 m, 1284 m, 1283 w, 1222 w, 1136 m, 1103 s, 990 s, 930 m, 864 m, 846 m, 826 m, 759 s, 725 s, 696 s, 650 m, 640 m, 619 w, 556 w, 546 w, 505 m.

The yield of IV was 0.170 g (67.5%) based on [{Cd(H_2_O)_4_(pfb)}^+^*_n_*·*n* (pfb)^−^]). Anal. Calc. for C_50_H_12.5_O_10_N_3.5_F_25_DyCd (%): C 38.2; H 0.8; N 3.1. Found: C 38.3; H 0.9; N 3.0. IR (ATR), ν/cm^–1^: 1651 m, 1627 w, 1589 s, 1569 m, 1519 w, 1488 m, 1450 m, 1433 m, 1384 s, 1348 m, 1285 m, 1288 m, 1222 w, 1139 w, 1099 w, 991 s, 925 w, 864 m, 845 m, 826 m, 755 s, 725 s, 701 m, 650 m, 638 m, 619 w, 553 w, 547 w, 505 m.


*[Tb_2_Cd_2_(phen)_2_(pfb)_10_](V).*


Complex V was synthesized as described above for compound III. The resulting reaction mixture was kept in a sealed vial at 75 °C. Colorless crystals suitable for X-ray diffraction studies that precipitated after 2 days were filtered off, washed with cold MeCN (T = 5 °C), and dried in air at 20 °C. The yield of V was 0.109 g (42.5%) based on [{Cd(H_2_O)_4_(pfb)}^+^*_n_*·*n* (pfb)^−^])). Anal. Calc. for C_94_H_16_O_20_N_4_F_50_Tb_2_Cd_2_ (%): C 37.6; H 0.5; N 1.9. Found: C 37.3; H 0.4; N 1.7.IR-spectra (ATR), ν/cm^–1^: 1723 m, 1698 m, 1651 m, 1613 s, 1520 s, 1490 s, 1430 m, 1390 s, 1316 m, 1227 m, 1140 w, 1103 s, 989 s, 929 m, 855 w, 844 s, 831 m, 761 m, 741 s, 725 s, 707 s, 642 m, 584 w, 504 w.


*[Ln_2_Zn_2_(phen)_2_(pfb)_10_]·4MeCN (Ln = Eu(VI), Gd(VII), Tb(VIII), Dy(IX)).*


0.314 g of H(pfb) (1.485 mmol) was added to as-precipitated Zn(OH)_2_ (0.075 g, 0.742 mmol) in 20 mL H_2_O. The reaction mixture was stirred at 75 °C for 2 h until complete dissolution of Zn(OH)_2_ and evaporated to dryness. The precipitate that formed was dissolved in 15 mL of MeCN, then [Ln_2_(pfb)_6_(H_2_O)_8_]·2H_2_O (0.371 mmol, Ln = Eu(VI), Gd(VII), Tb(VIII), Dy(IX)) was added to the resulting solution. The reaction mixture was kept in a sealed vial at room temperature. Colorless crystals suitable for X-ray diffraction studies that precipitated after 3 days were filtered off, washed with cold MeCN (T = 5 °C), and dried in air at 20 °C.

The yield of VI was 0.948 g (83.3%) based on Zn(OH)_2_. Anal. Calc. for C_102_H_28_O_20_N_8_F_50_Eu_2_Zn_2_ (%): C 39.9; H 0.9; N 3.7. Found: C 40.1; H 1.0; N 3.9. IR (ATR), ν/cm^–1^: 1723 m, 1698 m, 1651 m, 1613 s, 1521 s, 1490 s, 1427 m, 1396 s, 1316 m, 1225 m, 1147 w, 1103 s, 988 s, 932 m, 855 w, 844 s, 831 m, 761 m, 741 s, 725 s, 707 s, 640 m, 583 w, 507 w.

The yield of VII was 0.895 g (78.6%) based on Zn(OH)_2_. Anal. Calc. for C_103_H_32_O_22_N_8_F_50_Gd_2_Zn_2_ (%): C 39.3; H 1.0; N 3.6. Found: C 39.2; H 1.3; N 3.8. IR (ATR), ν/cm^–1^:1721 m, 1651 m, 1613 s, 1521 s, 1489 s, 1427 m, 1390 s, 1293 m, 1225 m, 1144 w, 1107 s, 1049 w, 990 s, 932 m, 854 w, 848 s, 827 m, 769 m, 755 m, 741 s, 725 s, 698 s, 635 m, 583 w, 507 w.

The yield of VIII was 0.651 g (58.5%) based on Zn(OH)_2_. Anal. Calc. for C_102_H_28_O_20_N_8_F_50_Tb_2_Zn_2_ (%): C 39.7; H 0.9; N 3.6. Found: C 39.5; H 0.8; N 3.7. IR (ATR), ν/cm^–1^:1721 m, 1651 m, 1613 s, 1521 s, 1489 s, 1427 m, 1392 s, 1293 m, 1225 m, 1147 w, 1107 s, 1047 w, 990 s, 932 m, 855 w, 848 s, 827 m, 769 m, 755 m, 741 s, 725 s, 697 s, 635 m, 583 w, 506 w.

The yield of IX was 0.700 g (61.1%) based on Zn(OH)_2_. Anal. Calc. for C_102_H_28_O_20_N_8_F_50_Dy_2_Zn_2_ (%): C 39.6; H 0.9; N 3.6. Found: C 39.8; H 0.8; N 3.5. IR (ATR), ν/cm^–1^:1720 w, 1652 m, 1613 s, 1521 s, 1490 s, 1427 m, 1392 s, 1296 m, 1225 w, 1149 w, 1106 s, 1047 m, 990 s, 930 w, 855 w, 848 m, 827 m, 769 m, 750 m, 738 m, 725 s, 699 s, 635 m, 583 w, 500 w.


*[Eu_2_Cd_2_(phen)_4_(pfb)_10_]^.^4MeCN (X).*


0.157 g of H(pfb) (0.742 mmol) was added to as-precipitated Zn(OH)_2_ (0.038 g, 0.371 mmol) in 20 mL H_2_O. The reaction mixture was stirred at 75 °C for 2 h until complete dissolution of Zn(OH)_2_ and evaporated to dryness. The precipitate that formed was dissolved in 40 mL of MeCN, then [{Cd(H_2_O)_4_(pfb)}^+^*_n_*·*n*(pfb)^−^] (0.224 g, 0.371 mmol), [Eu_2_(pfb)_6_(H_2_O)_8_]·2H_2_O (0.650 g, 0.742 mmol) and phen (0.267 g, 1.484 mmol) were added to the resulting solution. The reaction mixture was stirred for 5 min and left to evaporate slowly at 75 °C. Colorless crystals suitable for X-ray diffraction studies that precipitated after 5 days were filtered off, washed with cold MeCN (T = 5 °C), and dried in air at 20 °C. The yield of X was 0.150 g (22.3%) based on [{Cd(H_2_O)_4_(pfb)}^+^_n_•n(pfb)^-^]))). Anal. Calc. for C_126_H_44_O_20_N_12_F_50_Eu_2_Cd_2_ (%): C 42.9; H 1.3; N 4.8. Found: C 43.3; H 1.6; N 4.7. IR (ATR), ν/cm^–1^: 3659 w, 2988 w, 2901 w, 1647 m, 1591 m, 1518 m, 1486 s, 1427 s, 1383 s, 1294 w, 1224 w, 1137 w, 1105 m, 989 s, 930 m, 865 w, 846 m, 826 m, 747 s, 725 s, 696 m, 640 m, 582 m, 506 m, 484 w, 462 w.

## 5. Conclusions

A series of new M-Ln (M = Zn, Cd, Ln = Eu, Gd, Tb, Dy) heterometallic complexes with pentafluorobenzoic acid anions and 1,10-phenanthroline were synthesized. For the Cd-Ln complexes, the possibility of obtaining 1D polymers at RT and molecular complexes at 75 °C (for complex V as an example) was revealed, the composition of which differs only in the presence of acetonitrile solvate molecules for I–IV. At the same time, LnZn complexes form molecular compounds under similar conditions. The peculiarity of the formation of the polymeric LnCd structure is due to the lability of the carboxylate group, the functionality of which changes from μ_2_,η_2_ to μ_3_,η_2_ on passage from the molecular to polymeric form of the complex, and the accompanying enhancement of intramolecular π-π interactions between aromatic fragments (pfb anion and 1,10-phenanthroline) of neighboring tetranuclear fragments. Europium and terbium containing complexes exhibit bright metal-centered emission in the red and green regions, respectively. The luminescence efficiency is essentially limited by incomplete energy transfer from *d*-block to the emission center, but the presence of the phen ligand, the absence of H_2_O molecules (that are usually present in the composition of the lanthanide carboxylates) as well as high-frequency oscillating bonds achieved by the presence of *d*-block provide longer lifetimes of europium and terbium excited states and higher quantum yields (for the EuM complexes) than those of homonuclear europium and terbium perfluorobenzoates. The Gd(III) ions bound by carboxylate ligands in complexes II and VII cause very weak antiferromagnetic coupling between them. Complexes II-IV, VII and IX exhibit magnetic relaxation at helium temperatures in nonzero magnetic fields, but only for II, VII and IX the maxima on the curves χ^″^(ν) are in the frequency range of the equipment used. The magnetic relaxation of gadolinium compounds II and VII is associated with the QTM and Raman mechanisms. For dysprosium complex IX, magnetic relaxation is approximated by a sum of the Raman and direct relaxation mechanisms.

## Figures and Tables

**Figure 1 materials-13-05689-f001:**
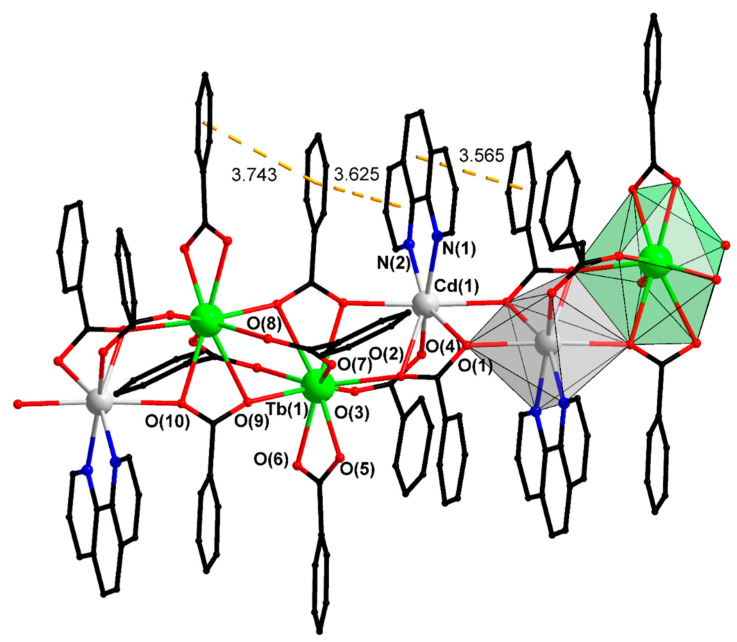
A fragment of polymeric chain of III. Hydrogen and fluorine atoms are omitted (dotted lines).

**Figure 2 materials-13-05689-f002:**
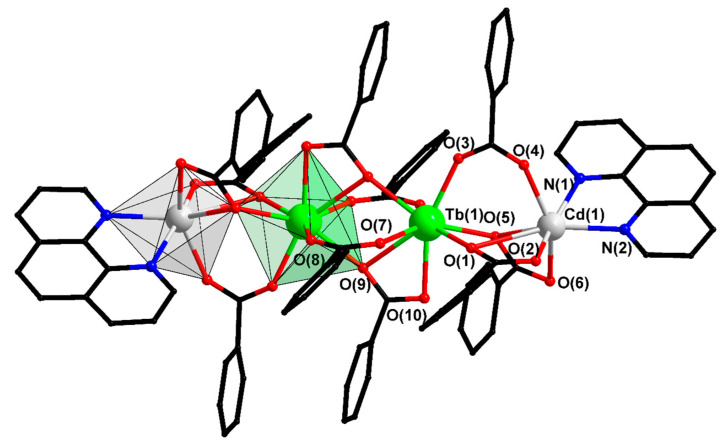
Structure of complex V. Hydrogen and fluorine atoms are omitted.

**Figure 3 materials-13-05689-f003:**
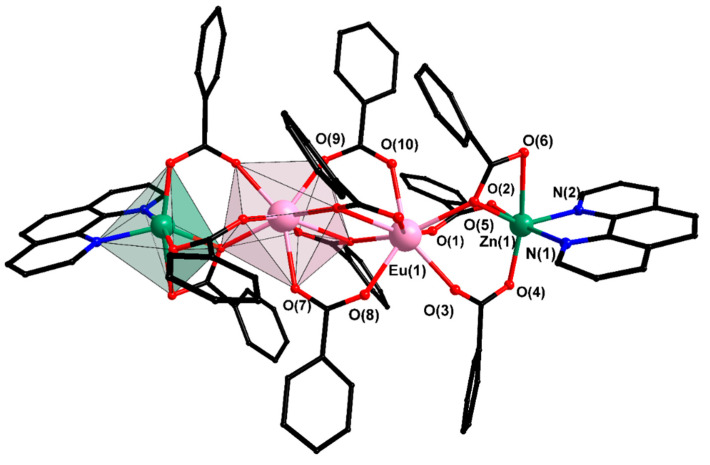
Structure of complex VI. Hydrogen and fluorine atoms are omitted.

**Figure 4 materials-13-05689-f004:**
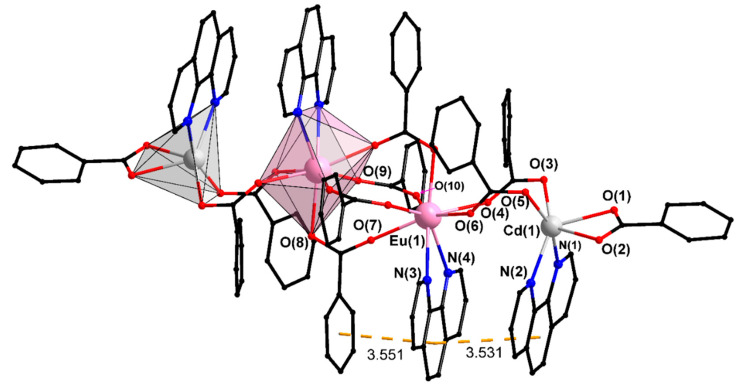
Structure of complex X. Hydrogen and fluorine atoms as well as solvate molecules are omitted.

**Figure 5 materials-13-05689-f005:**
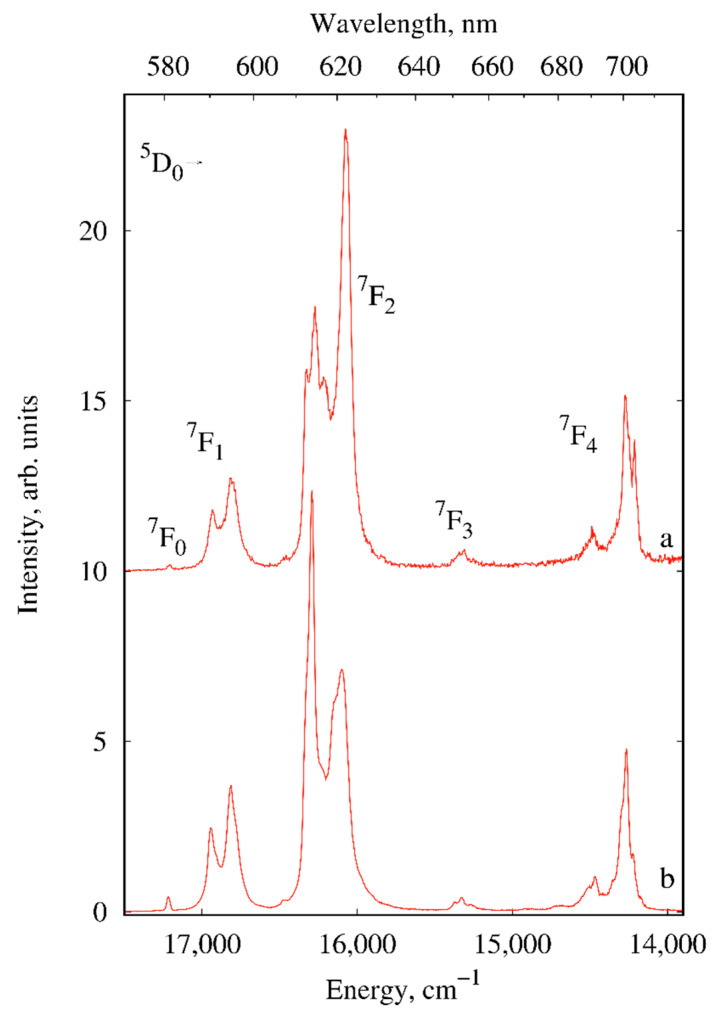
Luminescence spectra of I (**a**) and VI (**b**) (solid samples, λ_ex_ = 280 nm, *T* = 300 K).

**Figure 6 materials-13-05689-f006:**
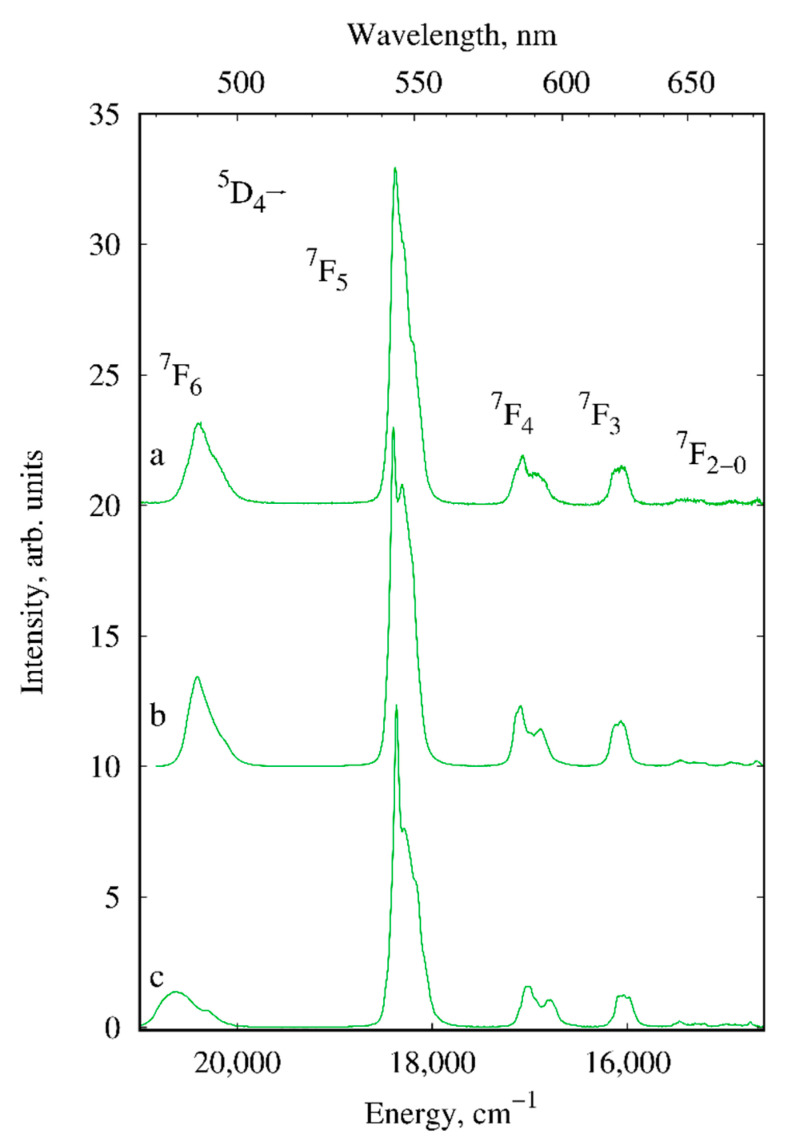
Luminescence spectra of III (**a**), V (**b**) and VIII (**c**) (solid samples, λ_ex_ = 280 nm, *T* = 300 K).

**Figure 7 materials-13-05689-f007:**
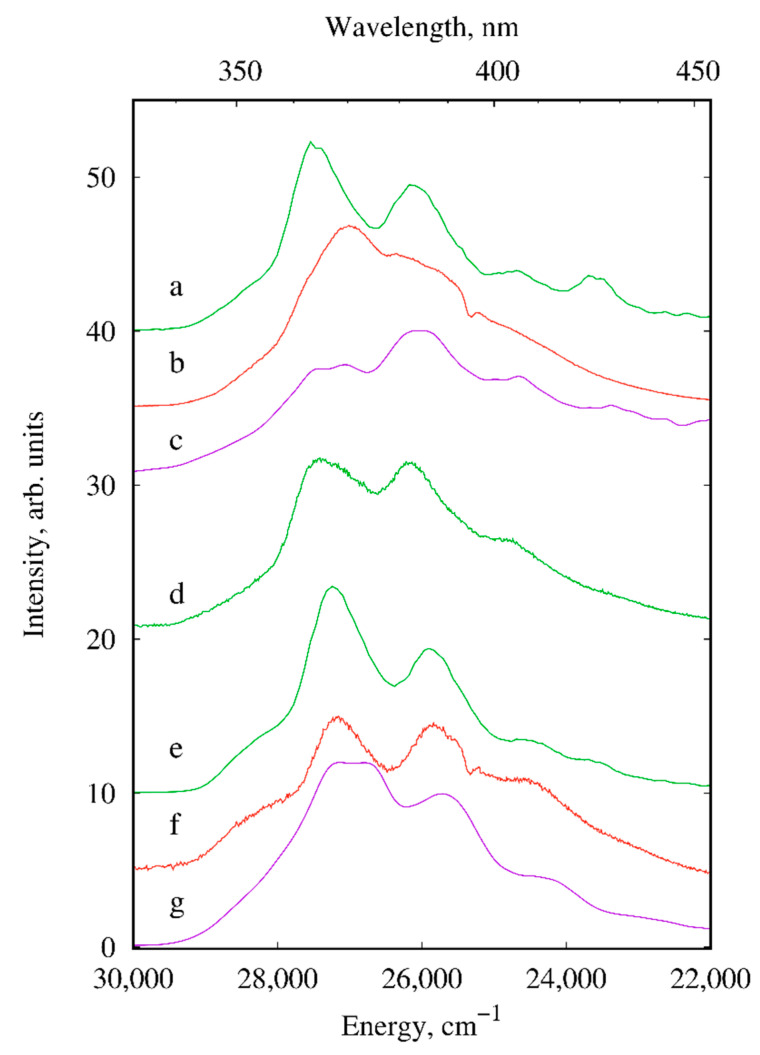
The d-block luminescence of III (**a**), I (**b**), II (**c**), V (**d**), VIII (**e**), VI (**f**) and VII (**g**) (solid samples, λ_ex_ = 280 nm, *T* = 300 K).

**Figure 8 materials-13-05689-f008:**
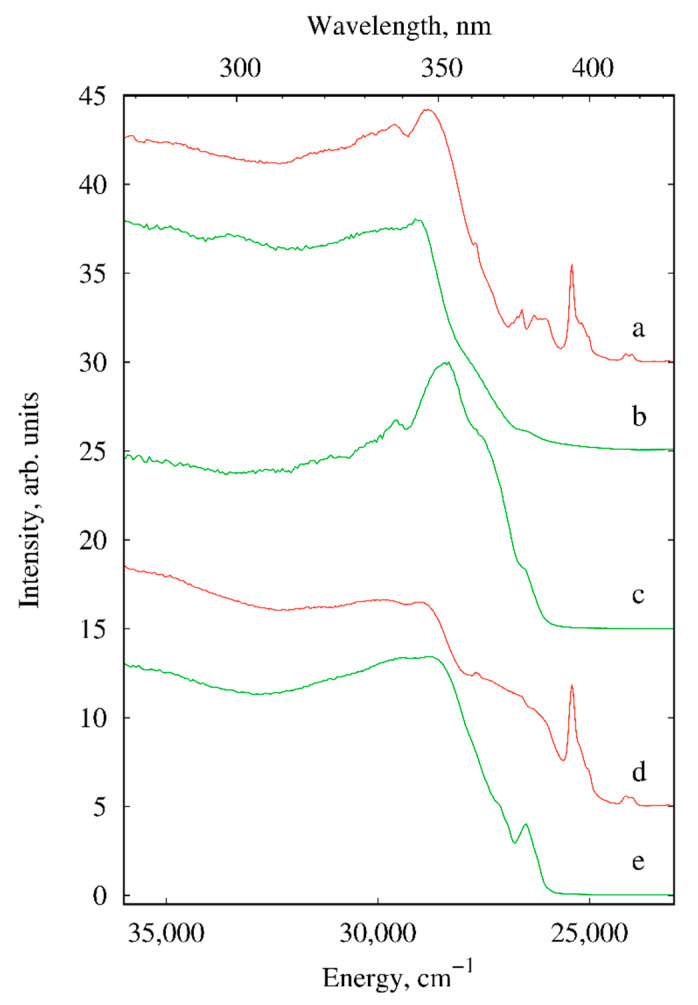
Excitation spectra of I (**a**), III (**b**), V (**c**), VI (**d**) and VIII (**e**) (solid samples, λ_em_ = 615 nm (for Eu^3+^ compounds) and 545 nm (for Tb^3+^ compounds), *T* = 300 K).

**Figure 9 materials-13-05689-f009:**
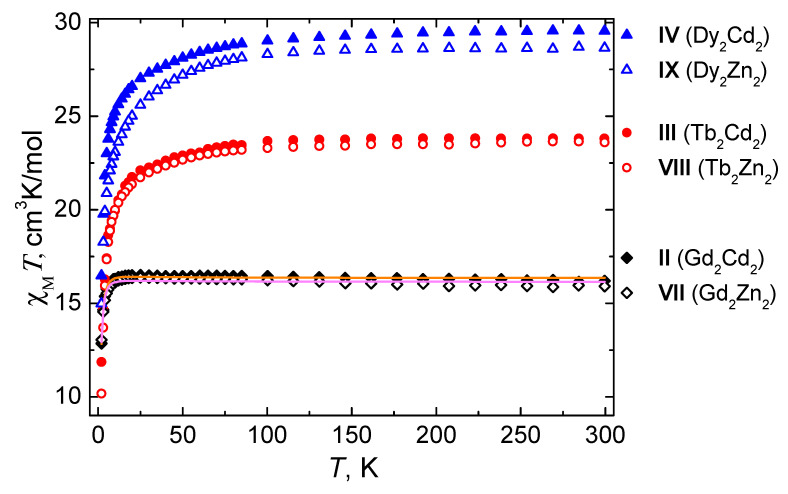
The χ_M_T vs. T dependencies for complexes II–IV (calculated per Ln_2_Cd_2_ unit) and VII–IX under 5000 Oe dc-magnetic field. Solid lines are the best-fit curves obtained for complexes II and VII using PHI program [43].

**Figure 10 materials-13-05689-f010:**
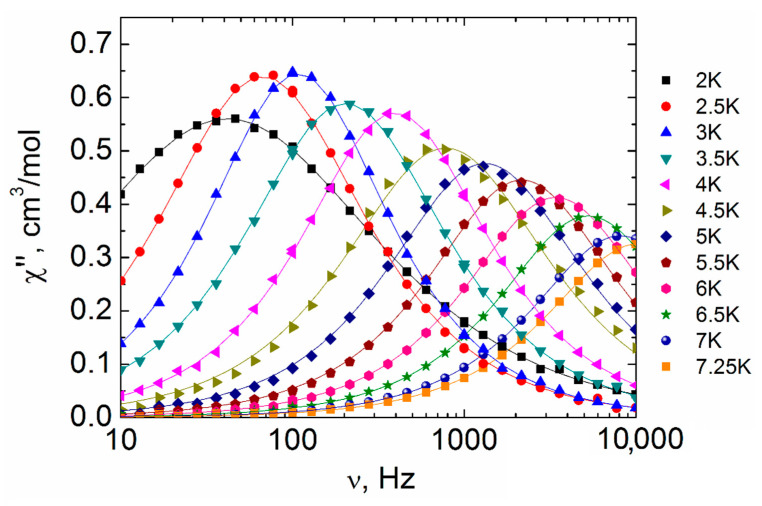
Frequency dependencies of the imaginary components of the ac-magnetic susceptibility for complex IX in the 2–7.25 K range taken under the optimal 1500 Oe dc-field. Solid lines represent fitting by the generalized Debye model.

**Figure 11 materials-13-05689-f011:**
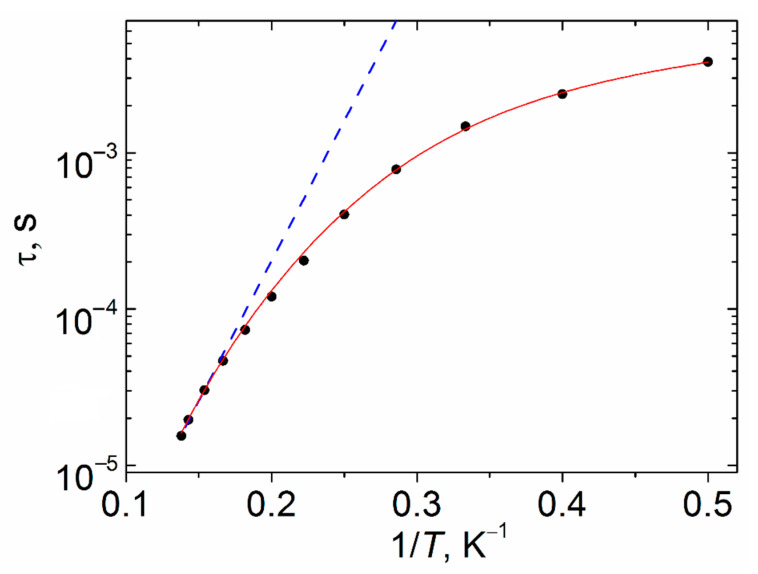
The τ vs. 1/T plot for complex IX under 1500 Oe field at *T* = 2 K. Blue dashed line represents fitting by the Orbach mechanism (Arrhenius equation). Solid red line represents fitting by the sum of direct and Raman relaxation mechanisms.

**Table 1 materials-13-05689-t001:** Selected bond lengths, the shortest interatomic distances *d* (Å) and angles *ω* (°) in structures I**–**V, X.

Compound/Parameter	I	II	III	IV	V	X
**Bond Lengths**	*d* (Å)					
**Cd–O (pfb)**	2.207(4)–2.468(5)	2.206(5)–2.480(5)	2.215(4)–2.469(4)	2.212(5)–2.485(6)	2.252(3)–2.405(3)	2.210(8)–2.457(10)
**Cd–N (phen)**	2.314(5), 2.320(5)	2.304(5), 2.317(5)	2.319(5), 2.324(5)	2.262(15)–2.44(2)	2.290(3), 2.345(3)	2.264(10), 2.320(11)
**Ln–O (pfb)**	2.370(3)–2.728(4)	2.364(4)–2.719(4)	2.350(4)–2.729(4)	2.329(4)–2.734(4)	2.314(3)–2.655(3)	2.325(7)–2.442(7)
**Ln–N (phen)**	-	-	-	-	-	2.602(9), 2.636(9)
**Interatomic distances**	*d* (Å)					
**Cd…Ln**	3.987(1)	3.971(2)	3.974(1)	3.964(1)	3.917(1)	4.453(1)
**Ln…Ln**	4.178(1)	4.173(2)	4.164(1)	4.153(1)	4.015(1)	4.408(1)
**Cd…Cd**	3.923(1)	3.948(2)	3.926(4)	3.940(1)	7.793(1)	6.782(19)
**Bond angles**	*ω* (°)					
**Cd–Ln–Ln**	115.25(3)	115.57(5)	115.23(3)	115.39(2)	171.62(1)	162.57(3)

**Table 2 materials-13-05689-t002:** Selected bond lengths, the shortest interatomic distances *d* (Å) and angles *ω* (°) in structures VI–IX.

Compound/Parameter	VI	VII	VIII	IX
**Bond Lengths**	*d* (Å)			
**Zn–O (pfb)**	2.037(4)–2.454(5)	2.037(2)–2.357(2)	2.037(3)–2.436(3)	2.028(3)–2.452(3)
**Zn–N (phen)**	2.088(5), 2.137(5)	2.108(3), 2.135(2)	2.091(3), 2.137(4)	2.088(3), 2.135(3)
**Ln–O (pfb)**	2.340(4)–2.608(4)	2.357(2)–2.572(2)	2.317(3)–2.607(3)	2.291(2)–2.576(2)
**Ln–O (H_2_O)**	-	2.426(2)	-	-
**Interatomic distances**	*d* (Å)			
**Zn…Ln**	3.857(1)	3.392(1)	3.864(1)	3.819(1)
**Ln…Ln**	3.986(1)	3.991(1)	3.965(1)	3.917(1)
**Zn…Zn**	7.783(1)	7.936(1)	7.828(1)	7.946(1)
**Bond angles**	*ω* (°)			
**Zn–Ln–Ln**	161.77	160.30	162.12	162.17

**Table 3 materials-13-05689-t003:** Radiative (A_rad_) and non-radiative (A_nrad_) rate constants, luminescence lifetimes (τ), intrinsic ($Q_{Ln}^{Ln}$) and overall ($Q_{Ln}^{L}$) quantum yields and sensitization efficiency (𝜂_sens_) of I, III, IV, VI and VIII.

Compound	A_rad_, s^−1^^*a*^	A_nrad_, s^−1^	τ, ms	$Q_{Ln}^{Ln},\;\%{\;}^{a}$	$Q_{Ln}^{L},\;\%{\;}^{b}$	𝜂_sens_, %
I (EuCd)	325	195	1.92 ± 0.05	62	36	58
VI (Eu_2_Zn_2_)	425	100	1.90 ± 0.05	81	41	51
[Eu(pfb)_3_(H_2_O)*_n_*] [13]	-	-	0.65	65	15	23
III (TbCd)	-	-	2.09 ± 0.06	-	63	-
V (Tb_2_Cd_2_)	-	-	1.95 ± 0.06	-	33	-
VIII (Tb_2_Zn_2_)	-	-	1.83 ± 0.05	-	45	-
[Tb(pfb)_3_(H_2_O)*_n_*] [13]	-	-	1.36	-	38	-

*a—*were measured at excitation wavelength 464 nm. *b—*were measured at excitation wavelength 280 nm.

**Table 4 materials-13-05689-t004:** The χ_M_T values of II–IV (calculated per Ln_2_Cd_2_ unit), VII–IX.

Complex	χ_M_T, cm^3^/mol K		
	Experimental (300 K)	Theor. (2Ln) [43]	Experimental (2 K)
II	16.19	15.76	12.86
III	23.80	23.64	11.87
IV	29.55	28.34	16.47
VII	15.91	15.76	13.04
VIII	23.59	23.64	10.17
IX	28.64	28.34	14.97

**Table 5 materials-13-05689-t005:** Approximation parameters of χ_M_T(T) dependences for complexes II and VII calculated by means of the PHI program [44].

Parameter	II	VII
	Value	
*g*	2.0375 ± 0.0008	2.024 ± 0.001
*J*, cm^−1^ (intramolecular interactions)	−0.074 ± 0.001	−0.070 ± 0.003
*zJ*, cm^−1^ (intermolecular interactions)	0.0204 ± 0.0006	0.020 ± 0.001
*R* ^2^	0.9968	0.9870

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
