# Peer review of "Molecular and Polymer Ln2M2 (Ln = Eu, Gd, Tb, Dy; M = Zn, Cd) Complexes with Pentafluorobenzoate Anions: The Role of Temperature and Stacking Effects in the Structure; Magnetic and Luminescent Properties"

_materials, 2020, doi:10.3390/ma13245689_

Round 1
Reviewer 1 Report
The paper introduced the results on the synthesis of LnZn and LnCd complexes with pentafluorobenzoic acid anions (Hpfb) and 1,10-phenantroline (phen) molecules. The study of the synthesis of complexes and the structural study of complexes were detail. It provided mean useful information.
Some content needed to be improved:
1.Please improve the quality of figures 5, 6, 7 and 8.
2.In Table 4. The χMT values of II-IV (calculated per Ln2Cd2 unit), VII-IX.
The term of exp. (300 K) was listed twice. Please revised it?
3.In the Table 5. Approximation parameters of χMT(T) dependences for complexes II and VII calculated by means of the PHI program [43].
Please provide the methods and equations to show that how to obtained these values, g, J, zJ and R2.
Author Response
Dear reviewers
thank you very much for your time and attention to our article. We hope that we have correctly considered your comments and remarks.
Reviewer 1
Some content needed to be improved:
1.Please improve the quality of figures 5, 6, 7 and 8.
Answer:
It was done.
2.In Table 4. The χMT values of II-IV (calculated per Ln2Cd2 unit), VII-IX. The term of exp. (300 K) was listed twice. Please revised it?
Answer:
The Table 4 was corrected.
3.In the Table 5. Approximation parameters of χMT(T) dependences for complexes II and VII calculated by means of the PHI program [43]. Please provide the methods and equations to show that how to obtained these values, g, J, zJ and R2.
Answer:
The necessary equations and calculation descriptions were added to the page 12.
Reviewer 2
- The grammar and stylistics in the paper should be improved. I provide some phrases/sentences to correct below (in 5), but the list is far to be complete.
Answer:
The text was checked additionally. Thank you for the comment.
- Please check L212, 5D0→7F5, 5D4?
Answer:
It was corrected.
- L120, angles and distances are provided without error values, please check throughout the manuscript
Answer:
The error values are given.
- L272-L274, I encourage the authors to delete/reformulate this part. It is not necessary to make assumptions based on “visual” appearance of the magnetic function, especially when this was revealed by correct analysis of the Gd compounds below. In Ln complexes, the exchange interactions are in most of the cases very weak, but not always. But in general, systems comprising interplay between large spin orbit coupling and exchange interaction might be tricky to treat.
Answer:
The sentences mentioned by reviewer were deleted.
- Other issues:
L44: “effect of quenching the fluorescence of the vibrational C-H bond” by instead of?
L70: “the interaction between”, the reaction between
L76-78, Sentence starting from “Isolation” is hard to follow
L98 and throughout the manuscript “one bridged carboxylate” bridging
L139 “is slightly differ by” slightly differ by
L157 and throughout the manuscript “hydrogen binding”, bonding
L290-291, this sentence is hard to follow, singular/plural mismatch
L338 “During the considere possible relaxation” hard to follow, typo
L362 “CBSD”, perhaps CSD or CCDC? Missing literature reference
Answer:
Your comments have been taken into account. Errors were corrected.
Reviewer 2 Report
The present article by Maxim A. Shmelev et al. entitled: “Molecular and polymer Ln2M2 (Ln = Eu, Gd, Tb, Dy; M = Zn, Cd) complexes with pentafluorobenzoate anions: the role of temperature and stacking effects in the structure; magnetic and luminescent properties “, describes the preparation, crystal structures, spectral and magnetic properties of large series of Ln- Cd(II)/Zn(II) complexes, which possess interesting di- or tetranuclear structures. The manuscript reports on interesting results and their discussion is sound. I find this manuscript to be interesting and suitable for publishing in Materials after minor revisions.
Comments:
- The grammar and stylistics in the paper should be improved. I provide some phrases/sentences to correct below (in 5), but the list is far to be complete.
- Please check L212, 5D0→7F5, 5D4?
- L120, angles and distances are provided without error values, please check throughout the manuscript
- L272-L274, I encourage the authors to delete/reformulate this part. It is not necessary to make assumptions based on “visual” appearance of the magnetic function, especially when this was revealed by correct analysis of the Gd compounds below. In Ln complexes, the exchange interactions are in most of the cases very weak, but not always. But in general, systems comprising interplay between large spin orbit coupling and exchange interaction might be tricky to treat.
- Other issues:
L44: “effect of quenching the fluorescence of the vibrational C-H bond” by instead of?
L70: “the interaction between”, the reaction between
L76-78, Sentence starting from “Isolation” is hard to follow
L98 and throughout the manuscript “one bridged carboxylate” bridging
L139 “is slightly differ by” slightly differ by
L157 and throughout the manuscript “hydrogen binding”, bonding
L290-291, this sentence is hard to follow, singular/plural mismatch
L338 “During the considere possible relaxation” hard to follow, typo
L362 “CBSD”, perhaps CSD or CCDC? Missing literature reference
Author Response

(The authors gave the same response as above.)
